# A Q fever outbreak associated to courier transport of pets

Eva Alonso[1], Domingo Eizaguirre[2], Idoia Lopez-Etxaniz[3], José Ignacio Olaizola[3], Blanca Ocabo[4], Jesús Felix Barandika[5], Isabel Jado[6], Raquel Álvarez-Alonso[5], Ana Hurtado[5], Ana Luisa García-Pérez[5]*

1 Department of Epidemiology, Subdirección de Salud Pública de Bizkaia, Gobierno Vasco, Bilbao, Bizkaia, Spain, 2 Department of Epidemiology, Subdirección de Salud Pública de Bizkaia, Gobierno Vasco, Vitoria-Gasteiz, Araba, Spain, 3 Department of Epidemiology, OSALAN-Instituto Vasco de Seguridad y Salud Laborales, Barakaldo, Bizkaia, Spain, 4 Department of Agriculture, Diputación Foral de Bizkaia, Bilbao, Bizkaia, Spain, 5 Animal Health Department, NEIKER- Instituto Vasco de Investigación y Desarrollo Agrario, Derio, Bizkaia, Spain, 6 National Microbiology Center, Instituto de Salud Carlos III, Majadahonda, Madrid, Spain

* agarcia@neiker.eus

**Data Availability Statement:** All relevant data are within the paper and its Supporting Information files. However, under the Data Protection laws, raw individual personal and clinical data extracted from the questionnaires cannot be provided since they

## Abstract

On August 3rd, 2017, a Q fever outbreak alert was issued at a courier company that in addition to urgent freight transport offered pet delivery services. The epidemiological investigation set the exposition period between June 1 and August 8. In this period, 180 workers from two operational platforms for parcel distribution located in two provinces of the Basque Country (Bizkaia and Araba) were exposed; 64 filled a questionnaire and provided blood samples for serological testing, resulting in 10 confirmed cases (15.6%) and six (9.4%) probable cases. Nine workers (8 confirmed and 1 probable) showed Q fever symptoms, including pneumonia (five cases), and required medical care services, including one hospital admission. The attack rate was 25% (16/64), being higher among workers that visited the Bizkaia platform. This suggested that the origin of the outbreak was in the Bizkaia platform, where animals in transit waited at a pet holding site until being moved to their destination. Environmental samples consisting on 19 surface dust and two aerosol samples were collected at the Bizkaia platform to investigate the presence of *C. burnetti* DNA. All dust samples were positive by real time PCR, the lowest Ct values being found in dust collected at the pet holding facilities, and therefore suggesting that contamination originated at the pet holding site. The genotype identified in dust was SNP1/MST13, one of the most commonly identified genotypes in goats and sheep in the Basque Country. During the exposure period, two deliveries of miniature goats were made, of which only one could be investigated and tested negative. Although the contamination source could not be unequivocally identified, transport of ruminants was banned at the company, and Q fever was included among the occupational-associated health risks.

represent sensitive information. However, any request for additional information will be considered and provided upon request to the Epidemiology Service of the Department of Public Health of the Basque Government (e-mail: epidemiologia-bizkaia@euskadi.eus).

**Funding:** This work was funded by the Spanish National Institute for Agricultural and Food Research and Technology (INIA) (RTA2017-00055-C02-00), the European Regional Development Funds (ERDF) and the Basque Government. RAA is beneficiary of a PhD contract funded by INIA (FPI-2015-014). The funders had no role in study design, data collection and interpretation, or the decision to submit the work for publication.

**Competing interests:** The authors have declared that no competing interests exist.

## Introduction

The intracellular bacterium *Coxiella burnetii* is the causative agent of Q fever in people and animals. It has been reported that the inhalation of 1 to 10 bacteria can cause disease in people. The most important symptoms are fever, pneumonia and / or hepatitis, though about 60% of infected people are asymptomatic [1]. *C. burnetii* is worldwide distributed, the only exception being New Zealand [1], and numerous outbreaks of Q fever have been reported. EFSA [2] compiled 39 outbreaks of Q fever in Europe between 1982 and 2007, 32 of them being related to small ruminants. This is due to the fact that Q fever infected farm animals shed millions of bacteria during abortion and parturition through different routes of excretion such as feces, uterine fluids, milk, and placentas [3]. However, pets [1,2], as well as wild species [4], may also carry *C. burnetii*. In general, outbreaks of human Q fever are related to contact with the animal reservoirs, which maintain the infection active and shed *C. burnetii* by different routes to the environment. Therefore farmers, veterinarians, shearers or slaughterhouse workers, among others, are those most commonly affected. However, outbreaks occasionally occur in urban nuclei far from the source of infection when weather conditions favor wind transport of *C. burnetii* [5–8]. Moreover, *C. burnetii* can also contaminate materials and be transported from one place to another giving rise to unexpected outbreaks, and making the epidemiological investigation challenging. Environmental sampling proved very useful in tracing back recent outbreaks that occurred in our region at workplaces without apparent contact with animals [9,10]. In addition, the DNA obtained from dust allowed genotyping of the involved isolate and helped to identify the probable source of infection. In this study we present the investigation of an outbreak of Q fever in an express transport company authorized for the transport of pets.

## Material and methods

### Case presentation

The outbreak occurred in the Basque Country (northern Spain), a region divided into three provinces: Araba, Bizkaia and Gipuzkoa. On August 3, 2017, the Microbiology Laboratory of *Hospital Universitario de Cruces* (Bizkaia) informed the Epidemiology Surveillance Unit of the province of a possible Q fever outbreak at a courier company located in Bizkaia. The company confirmed that in July several workers of the company branch in Bizkaia were on sick leave. Also in July, another two possible cases of Q fever were reported by the Health Service from a nearby Spanish region (Castilla-León) in two workers of the Araba branch of the same company (60 km away from the Bizkaia platform) who lived in Castilla-León. Since the company was a licensed pet transporter, transported animals were suspected as a possible source of the Q fever outbreak. Under these circumstances, by August 3 the facility where animals were kept had already been cleaned and disinfected, and transport of animals had been suspended.

### Courier company description

The outbreak occurred in a courier company that in addition to urgent freight transport of parcels offered pet delivery services. The company is organized in several sections: reception and delivery offices where clients place the parcels; operational platforms for parcel distribution; and, transport services. Only workers at operational platforms are employees of the courier company, whereas reception and delivery offices are franchises and drivers are self-employed.

Pet delivery services offered by the company included transport and delivery of all type of animals, the only restriction being the maximum weight allowed. Pet delivery needs to comply with certain requirements of animal health and welfare. Notably, animals ought to be healthy

and health status be accredited by an official health certificate that complies with regulations at origin and destination. Besides, animals cannot be pregnant or not having recently given birth when travelling. Transport routes include delivery to the local platform by the collection site, transfer to the central platform in Madrid, where a veterinarian visually examines the animal, and transport to the local platform at destination. In each platform animals are located in specific pet holding facilities while waiting (less than 1 hour) to be transported. Maximum transit times are 24 hours. About 6–7 animals are delivered each day. Transported animals are mostly dogs and cats, while other animals like miniature sheep and goats, birds or rabbits are delivered once per month on average.

## Epidemiological investigation

Once the Epidemiology Surveillance Unit was informed of the suspected Q fever outbreak, a multidisciplinary group that included microbiologists, veterinarians, occupational health technicians and epidemiologists was gathered to investigate the infection source, monitor the exposed workers and design control measures.

On August 9, the Epidemiology Surveillance Unit and the Occupational Health Authority (OSALAN) met with the company management, the doctor and the occupational health technicians to collect information on the outbreak circumstances. The company informed that on July 16 they became aware of several workers from the Bizkaia and Araba branches reporting health problems compatible with Q fever starting on July 4. The following day, July 17, transport of ruminants was suspended at affected branches. That same day, pet holding facilities at the Bizkaia platform were cleaned with soap and bleach by the platform workers, who did not use the necessary protection equipment (*eg.* respiratory protection masks). Later, on July 22, a specialized cleaning company was subcontracted to clean and disinfect the pet holding facilities at the Bizkaia and Araba platforms, as well as the van used to transport animals. On July 24, transport of ruminants was banned for all transport routes.

Taking into account the incubation period of Q fever infection (2–3 weeks) and the onset date of the clinical signs in the affected patients, the exposure period was defined as that extended between June 1 and August 8. All workers that had any contact with either of the two affected platforms during that period, independently of their contractual relationship with the company, were considered exposed. Two meetings were held, one on each platform, to inform the workers, and conduct an epidemiological questionnaire and a serological study among workers who volunteered to participate. Workers unable attend to the meetings were approached by the Epidemiological Surveillance Unit by phone and asked to complete the questionnaire (S1 Appendix) and those who agreed to also provide blood samples. The workers' general practitioners (GP) and local hospitals were informed of the outbreak.

## Laboratory methods

Serological analyses were carried out at different hospitals and health prevention services. Two blood samples were obtained 3 to 4 weeks apart for serological determination of *C. burnetii* phase II IgM and IgG antibodies to evaluate seroconversion in exposed workers. Laboratory results were considered positive when seroconversion was observed; laboratory results were considered doubtful when only one blood sample was available and an indirect immunofluorescence antibody test (IFAT) titers for phase II IgG were >1/128 and for phase II IgM of >1/256.

Since ruminants are considered the main source of human Q fever infection in the study area, the investigation of potential infection sources focused on small ruminants that had travelled through either of the affected platforms (Bizkaia and/or Araba) during the exposure period (June 1—August 8). Blood samples (4 mL) were collected from animals to investigate

the presence of antibodies against *C. burnetii* in their blood using a commercial indirect ELISA test (LSIVET Ruminant Milk/Serum Q Fever kit; Thermo Fisher Scientific).

Environmental samples (air and dust) were collected at different locations of the Bizkaia platform on September 15. Two aerosol samples were taken with an air sampler (Airport MD8, Sartorius) inside the platform of Bizkaia. Nineteen surface dust samples were collected using cotton swabs in areas with accumulated dust that included the pet holding facilities and other sites in the company premises. After DNA extraction, samples were analyzed by Real Time PCR (RTi-PCR) [11]. Samples with a positive RTi-PCR result and a Ct value below 31 were genotyped by multispacer sequence typing (MST) and a 10-loci single-nucleotide polymorphism (SNP) discrimination using RTi-PCR as described elsewhere [12,13].

## Case definition

A confirmed human case was defined as a worker who had any contact with the Bizkaia and/or Araba platforms of the courier company between June 1 and August 8, 2017, independently of their contractual relationship with the company, and showed seroconversion in laboratory analyses, with or without clinical symptoms (fever, pneumonia and/or hepatitis). Probable cases included those exposed workers with mild clinical symptoms who tested serology positive without seroconversion or produced doubtful laboratory results (see Laboratory methods).

## Statistical analyses

Associations between personnel risk factors and data collected in the epidemiological investigation were analyzed by Chi squared test (categorical variables) or by Student's t–test (numerical and dichotomous variables) using SPSS Statistic 21. Attack rates were assessed by Mantel-Haenszel Chi squared test using Epi Info 7 and Odds Ratios were assessed by exact test using Stata 12.1.

## Ethical considerations

Since outbreaks are routinely investigated according to the Public Health services' ethical guidelines to ensure patients safety, this study did not require additional ethical approval. Written informed consent was obtained from the workers for blood sample collection and for filling a questionnaire, which included personal data collection following legal regulations (Ley Orgánica 15/1995). Data analysis was performed on an anonymized dataset.

Blood samples were taken directly from the jugular vein of the goats following Spanish ethical guidelines and animal welfare regulations (Real Decreto 53/2013) after obtaining informed consent from the flock owner. According to this regulation, extraction of blood samples from livestock in this type of studies is considered routine veterinary clinical practice and does not require ethical approval. Samples were collected by the highest authority in animal research, welfare and ethics in the regions, i.e., veterinarians of the local Animal Health and Welfare Authorities (Servicio de Ganadería, Diputación Foral de Bizkaia and Servicio de Sanidad y Producción Animal de Lugo, Xunta de Galicia), responsible of livestock sanitation and studies of zoonotic outbreaks.

# Results

## Epidemiological investigation

A total of 180 workers from 29 different organizations (including self-employees) were considered to be exposed to the infection and were informed of the outbreak. Of them, 36 lived in other regions and their cases were referred to their local Health Services. Sixty-four of the remaining 144 workers agreed to fill a questionnaire to investigate risk factors of exposure and

**Table 1. Summary of the results obtained in the epidemiological questionnaire according to variables and case definition.**

| Epidemiological data | | no. (%) | | |
|---|---|---|---|---|
| | | Confirmed cases (N = 10) | Probable cases (N = 6) | Non-case (N = 48) |
| **Sex** | Male | 9 (90.0) | 6 (100.0) | 44 (91.7) |
| | Female | 1 (10.0) | - | 4 (8.3) |
| **Age** | Mean | 45.8 | 36.5 | 38.5 |
| | Median | 45.5 | 35.0 | 36.5 |
| | Min—Max | 33–60 | 25–54 | 22–58 |
| **Symptoms** | Any symptom | 8 (80.0) | 1 (16.7) | 7 (14.6) [a] |
| | Fever / flu-like | 8 (80.0) | 1 (16.7) | 7 (14.6) [a] |
| | Pneumonia | 5 (50.0) | - | 1 (2.1) [a] |
| **Medical care** | Hospitalization | 1 (10.0) | - | - |
| | Emergency services | 6 (60.0) | - | 6 (12.5) [a] |
| | Primary health care services | 1 (10.0) | 1 (16.7) | 1 (2.2) [a] |
| **Living in a rural setting** | | 2 (20.0) | 1 (16.7) | 3 (6.3) |
| **Smoker** | | 9 (90.0) | 2 (40.0) | 18 (42.9) |
| **Contact with animals (at work)** | | 9 (90.0) | 3 (60.0) | 36 (75.0) |
| **Working at reception and/or transport services** | | 7 (87.5) | 3 (75.0) | 26 (59.1) |
| **Involved in cleaning operations** | | 1 (16.7) | 1 (33.3) | 5 (13.9) |

[a] these workers had compatible symptoms during the exposure period but they did not meet the laboratory criteria of Q fever.

to provide blood samples for serological testing. Ten of them (15.6%) were identified as confirmed cases and another six (9.4%) met the definition of probable case; the remaining 48 were regarded as non-cases. Nine workers (8 confirmed and 1 probable) showed symptoms compatible with Q fever, including five cases with pneumonia (Table 1). They all required medical care services, including one who needed hospital admission. Another 7 workers had compatible symptoms (including one case of pneumonia) in the outbreak period but none of them met the laboratory criteria of Q fever, and they were thus considered as non-case. The attack rate among the investigated workers was 25% (16/64). Although it was difficult to precisely know if workers visited just one or both of the affected platforms, attack rate at the Bizkaia platform was estimated to be 28.3% (13/46) and 16.7% (3/18) at the Araba platform, all of the latter being asymptomatic. Serological results compiled from workers are shown in S1 Table.

Median and mean age in confirmed cases did not significantly differ from non-affected workers. Percentage of smokers was higher in confirmed and probable cases than in non-cases ($p = 0.043$). Proportion of workers that had contact with animals at work was higher among confirmed cases, but differences were non-significant ($p>0.05$). Likewise, attack rate among the personnel involved in cleaning operations was non-significantly higher. No significant differences in incidence were observed between workers living in a rural or urban setting.

The epidemic curve representing the progression of illnesses onset in confirmed and probable cases is shown in Fig 1. First symptoms were recorded on July 5, and extended until August 2, with most cases concentrating during the first two weeks of July; only two cases delayed to August 1 and 2, after cleaning operations at the Bizkaia platform and the ban on animal transport was implemented.

## Outbreak source investigation: Animal and environmental study findings

Among all animal delivery services carried out between June 1 and August 8, two were considered the most probable sources of the infection. Delivery 1 involved two miniature goats

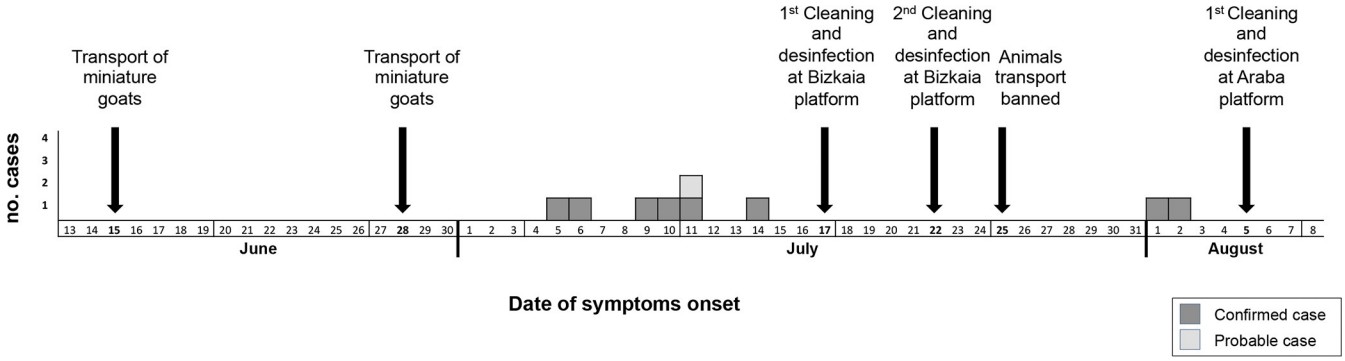

**Fig 1. Schematic representation of the main events associated to the outbreak.**

travelling on June 28 from Bizkaia to Orense. Travelling route was as follows: reception at the Bizkaia platform–Araba platform (animals did not leave the van)–central headquarters at Madrid (360 km from Araba platform), where a veterinarian inspected the animals and they were then transferred to a different van–reception office and delivery at Orense. The farm where these miniature goats originated from was inspected by Animal Authority Services. Blood samples were collected from five miniature goats and seven sheep for *C. burnetii* ELISA test (August 23), all of them being negative. The two suspected miniature goats also tested negative in the ELISA test when investigated at destination (August 25). Delivery 2 took place three weeks before the onset of the outbreak. It also involved the transport of miniature goats but animals only passed through the Bizkaia platform while travelling from Cantabria to Valencia. Since animals did not travel through the Araba platform, and considering the time elapsed since the outbreak and the difficulties associated to the several regions involved, these animals were not investigated.

Being the suspected animal source negative, an environmental study was set up to investigate the presence of *C. burnetii* at the company premises surrounding the pet holding facilities of the Bizkaia platform, where the higher number of cases occurred. Although by this time the Bizkaia platform had been cleaned and disinfected twice, on September 15, environmental samples (19 surface dust and two aerosols) were collected at different sites of the Bizkaia platform premises. Special attention was paid in collecting samples from sites with accumulated dust and dirt, both inside and outside the pet holding facilities as well as at different surfaces within the company premises (see Fig 2). Real time PCR results were weak positive (Ct>35) for the aerosol samples but all dust samples were positive (Ct≤35), with the lowest Ct values for samples collected at the pet holding facilities (Ct 23.9 and 27.9) (Fig 2). Specifically, the lowest Ct value (23.9) was found for a dust sample collected from the rails of the sliding doors of the pet holding site, where cleaning operations did not reach. These results suggested that contamination had originated there and then spread through the platform premises. The two samples collected at the pet holding site were subjected to genotyping but only one (Ct = 23.9) could be completely genotyped as SNP1 and MST13.

## Undertaken control measures

As mentioned above, on July 24, transport of ruminants was banned for all transport routes. After the initial cleaning operations at the Araba platform in July, new cleaning and disinfection were performed on August 5 and August 19 using 1% Virkon® S (Bayer Hispania S.L., Barcelona, Spain). At the Bizkaia platform, a third and more stringent cleaning and disinfection procedure was implemented in September 15 upon receipt of the results from the

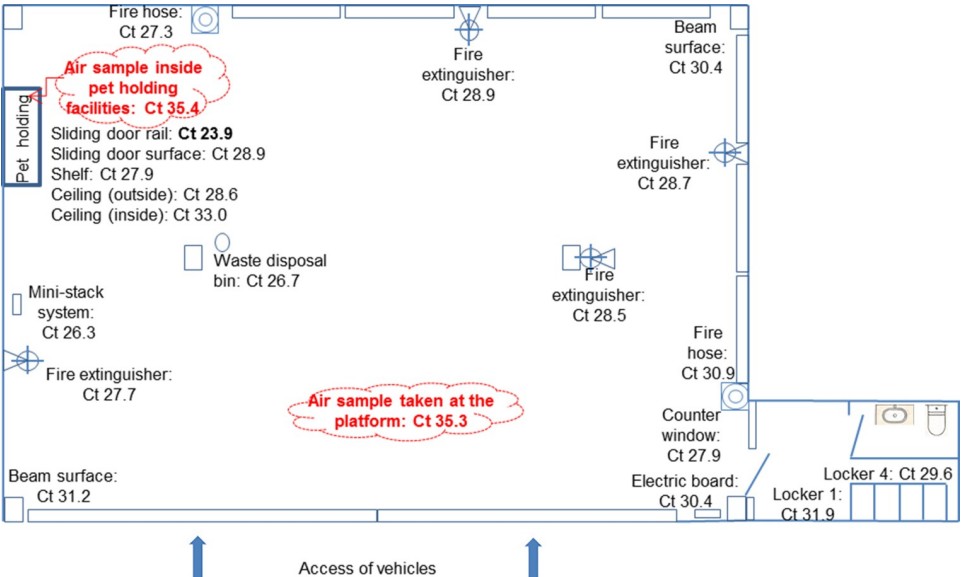

**Fig 2. Schematic representation of the Bizkaia platform (2,500 m$^2$) along with the sampling sites and their corresponding *C. burnetii* Real Time PCR results (Ct values).**

environmental study that confirmed *C. burnetii* contamination. It involved exhaustive cleaning and disinfection of the pet holding facilities (inside and outside) using wet methods (avoiding any procedures that involved dust formation such as sweeping or aspiration), as well as cleaning and disinfection with water and Virkon® S (Bayer Hispania S.L., Barcelona, Spain) of floors, banisters and any other dust accumulating sites. Differently to the operations in July, now the necessary protection equipment, *i.e.* respiratory protection masks (filtering face pieces class 3-FFP3), gloves and goggles, were used during cleaning and disinfection.

Prevention Services of the affected companies were asked to follow-up confirmed and probable cases in order to assess possible evolution to chronic Q fever. For self-employees monitoring would be carried out by their GPs.

## Discussion

A Q fever outbreak was declared in August 2017 at a courier company. The epidemiological investigation of the outbreak was hampered by the structure of the service that included not only personnel directly employed by the courier company but also workers from franchises and self-employed drivers. This situation hindered the identification of the personnel outside the platforms exposed to the infection source. In addition, workers had their residence in different regions and therefore the collaboration of different Health Services was required. Consequently, not all workers were included in the epidemiological survey. Despite these difficulties, the epidemiological and laboratory studies identified Q fever in 25% of the investigated workers. Pneumonia was the most relevant clinical sign and the cause of hospitalization of one patient. These results support previous studies which also identified pneumonia as the main clinical manifestation associated to other Q fever outbreaks recently reported in the Basque Country [9,10,14], unlike other Spanish regions where hepatitis is the most common clinical presentation [15]. In this sense, the Basque Country has been reported to have the highest rate of Q fever-associated pneumonia in Spain [15,16] and Europe as reviewed elsewhere [17].

Two deliveries of small ruminants (miniature goats) that took place within three weeks prior to the onset of the outbreak were considered as the most probable source of the outbreak. The first one, which took place three weeks before the outbreak (June 15), unfortunately could not be investigated. The second delivery, on June 28, involved the transport of two miniature goats from Bizkaia to Orense passing through both affected platforms. After serological investigation these animals did not show antibodies against *C. burnetii*, and neither did other animals from their flock of origin. It is nevertheless known that serology is not a good indicator of infection status since a percentage of *C. burnetii* infected ruminants do not seroconvert [18]. The high values of *C. burnetii* DNA detected by Real Time PCR on dust samples collected at the pet holding facilities of the Bizkaia platform clearly indicated that contamination originated there. The hypothesis that small ruminants were the infection source was supported by the *C. burnetii* genotype identified from dust samples (SNP1 / MST13), which corresponds to a type already detected from sheep [19] and goats [14] in the region. Although other pets like cats and dogs can also carry *C. burnetii* [20], genotype SNP1 / MST13 points at small ruminants as the source. In the study region cats and dogs have never been associated to Q fever outbreaks, but outbreaks were reported elsewhere after cats or dogs gave birth [21,22]. Although cats and dogs cannot be fully ruled out as a possible source of the outbreak, transport of parturient animals is not allowed. Considering all these, the contamination source could not be unequivocally identified. In the case of small ruminants, infected animals are known to shed *C. burnetii* through feces for several weeks after parturition or abortion [14,23]. Fecal contamination could have been the origin of the spread of high level of bacteria to the environment. Before transport, animals are kept for a while (less than 1h, according to the information provided by the company) in pet holding facilities, a place that workers frequently visited. In fact, about 72.6% of the interviewed workers confirmed direct or indirect contact with the animals at the platforms. Moreover, the air ventilation system of the pet holding facilities expelled air towards the platform. This might have been the main contamination route since *C. burnetii* DNA was detected all over the platform premises, the lowest PCR Ct values being found at the pet holding facilities. Although detection of *C. burnetii* DNA in dust samples collected far after the onset of the clinical cases and even after cleaning and disinfection does not prove that the bacteria were viable, it does suggest that contamination originated there. From there, travel containers, transport vans or workers travelling from Bizkaia to Araba could have acted as vehicles for *C. burnetii* contamination of the Araba platform, where three workers seroconverted but did not show clinical signs of infection. This would explain why attack rate was higher among workers that visited the Bizkaia platform, and those in closer contact with the transported animals.

Until recently, Q fever was considered an occupational disease linked only to the livestock sector. This outbreak highlights the risk of Q fever infection in other working settings where contact with potential animal reservoirs also occurs. Although animal delivery was included among the services offered by the company, Q fever was not considered among the occupational-associated health risks. In fact, even though animals need an official health certificate to travel, the analysis of *C. burnetii* antibodies is not legally requested, not even for domestic ruminants as occurs for example for *Brucella* spp. or tuberculosis in sheep and goats. This study highlighted the need to revise policies associated to animal health requirements for transport and consider the inclusion of laboratory tests (serology and PCR) to rule out *C. burnetii* infection.

This outbreak adds up to other cases and outbreaks where *C. burnetii* has been accidentally transported to environments far from livestock [5,8,24]. In such cases, outbreak investigations are even greater challenges that need rapid and coordinated One Health approaches. Here, expertise in human, animal and environmental health gathered together to collect epidemiological data and produce laboratory results to investigate the infection source and implement

control and prevention measures. Exhaustive cleaning and disinfection procedures were implemented at both affected platforms after the epidemiological outbreak investigation. Transport of ruminants was banned for all transport routes at the company, and Q fever was included among the company occupational-associated health risks.

## Supporting information

**S1 Appendix. Questionnaire used in the epidemiological investigation.**
(DOCX)

**S1 Table. Serological data compiled from workers.**
(XLSX)

## Acknowledgments

We would like to thank all the clinicians and laboratory staff of the Microbiology Services of Osakidetza, the Prevention Services and the Mutual Insurance for Occupational Accidents involved in the management of the outbreak. We are also grateful to the staff of the platform of Bizkaia during the visits and samplings.

## Author Contributions

**Conceptualization:** Ana Hurtado.

**Data curation:** Domingo Eizaguirre, Idoia Lopez-Etxaniz, José Ignacio Olaizola, Blanca Ocabo.

**Formal analysis:** Eva Alonso, Isabel Jado, Raquel Álvarez-Alonso.

**Investigation:** Eva Alonso, Domingo Eizaguirre, Idoia Lopez-Etxaniz, José Ignacio Olaizola, Blanca Ocabo.

**Methodology:** Eva Alonso, Jesús Felix Barandika, Isabel Jado, Ana Luisa García-Pérez.

**Supervision:** Eva Alonso, Ana Hurtado, Ana Luisa García-Pérez.

**Writing – original draft:** Eva Alonso, Ana Hurtado, Ana Luisa García-Pérez.

**Writing – review & editing:** Ana Hurtado, Ana Luisa García-Pérez.

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
