## [Decision Letter · Decision Letter 0]

21 Aug 2019

PONE-D-19-18129

A Q fever outbreak associated to courier transport of pets

PLOS ONE

Dear Dr. García-Pérez,

Thank you very much for submitting your manuscript "A Q fever outbreak associated to courier transport of pets" (#PONE-D-19-18129) for review by PLOS ONE. As with all papers submitted to the journal, your manuscript was fully evaluated by academic editor (myself) and by independent peer reviewers. The reviewers appreciated the attention to an important health topic, but they raised substantial concerns about the paper that must be addressed before this manuscript can be accurately assessed for meeting the PLOS ONE criteria. Therefore, if you feel these issues can be adequately addressed, we invite you to submit a revised version of the manuscript that addresses the points raised during the review process. We can’t, of course, promise publication at that time.

We would appreciate receiving your revised manuscript by Oct 05 2019 11:59PM. To enhance the reproducibility of your results, we recommend that if applicable you deposit your laboratory protocols in protocols.io, where a protocol can be assigned its own identifier (DOI) such that it can be cited independently in the future. For instructions see: http://journals.plos.org/plosone/s/submission-guidelines#loc-laboratory-protocols

We look forward to receiving your revised manuscript.

Kind regards,

Abdallah M. Samy, PhD

Academic Editor

PLOS ONE

**Additional Editor Comments:**

I invited and received three reviews for your manuscript. All reviews raised some substantial concerns about your manuscript as it currently stands. I read through their comments and found that they coincided on several points, and that their reviews were uniformly solid and detailed. I read the manuscript myself, and I must say that I coincide with the reviewers' points entirely. As such, I would recommend “major revision”. I would kindly ask you to go through all comments raised by each reviewer and address them properly before sending a revised version of this manuscript. Please check all PLOS ONE style requirements available via https://journals.plos.org/plosone/s/submission-guidelines before submitting the revised version.

**Journal Requirements:**

2. In your Methods, please state the volume of the blood samples collected for use in your study."

**Reviewers' comments:**

Reviewer's Responses to Questions

**Comments to the Author**

1. Is the manuscript technically sound, and do the data support the conclusions?

Reviewer #1: Yes

Reviewer #2: Partly

Reviewer #3: No

2. Has the statistical analysis been performed appropriately and rigorously? 

Reviewer #1: I Don't Know

Reviewer #2: Yes

Reviewer #3: Yes

3. Have the authors made all data underlying the findings in their manuscript fully available?

Reviewer #1: Yes

Reviewer #2: Yes

Reviewer #3: Yes

4. Is the manuscript presented in an intelligible fashion and written in standard English?

Reviewer #1: No

Reviewer #2: Yes

Reviewer #3: No

5. Review Comments to the Author

Reviewer #1: Although I believe that the authors have investigated the outbreak with care, I do not believe that the article has any new or illuminating insights. It is not at all surprising that an outbreak of Q fever occurred in a business that transported goats and sheep. The association of Q fever with these ruminants was established decades ago, and the ability of Coxiella burnetii to survive in dust for long periods is also well-known. The clinical features of disease in the infected patients in this study are consistent with previous reports, with no novel elements noted.

Reviewer #2: AUTHORS

Manuscript Number: PONE-D-19-18129

Manuscript Title: A Q fever outbreak associated to courier transport of pets

This is an outbreak report of Q fever in an express transport company linked to the transport of caprine, that had epidemic characteristics. It is an interesting and valid text that has an impact on the Public Health approach of this zoonosis. However I have major concerns that need to be addressed before acceptance.

1. Line 78: “possible cases of Q fever in two workers of the Araba branch of the same company who lived in Castilla-León.” What do you mean by Bizkaia and Araba branches? Are these cities? How far apart? How is this connected to Castilla-Leon? I believe this sentence would gain if rephrased for clarification.

2. Line 144: “masks). Later on, on July 22, a specialized” delete “on”

3. No environmental samples were taken from Araba and no link was found between these two branches that could justify a common cause (for example, same animal provider), other than occurring on the same dates. Additionally, molecular evidence exists only from Bizkaia and cannot confirm identic origins with Araba. Authors need to clarify why they considered this the same (multicentric?) outbreak.

4. Authors suspect that two miniature goats travelling on June 28 from Bizkaia to Orense were the source of infection: reception at the Bizkaia platform – Araba platform (animals did not leave the van) – central headquarters at Madrid, where animals were transferred to another vehicle. How did these two animals shed C. burnettii in Araba (even not being handled), but in Madrid, where they were transferred and handled, no humans cases were reported? Although possible this is not likely, hence I would suggest to discuss these drawbacks.

Reviewer #3: “A Q fever outbreak associated to courier transport of pets” (PONE-D-19-18129) for PLOS ONE

Reviewer’s comments

This manuscript represents a detailed account of an outbreak investigation associated with transportation of pets via a courier transport service. While it is an important study, there are considerable shortcomings related to lack of detail in the methods concerning information collected in the questionnaire, participant recruitment, the counting of cases and non-cases is recorded differently in the abstract and in results, and there is lack of clarity in some sections.

Title

The title is misleading as it indicates that the outbreak was associated with pets when in fact the source of infection was the carrying of goats – transportation of livestock/ruminants has a different risk to carrying domestic pets.

Abstract

Line 26 – I’m confused as about the term parcel distribution – this needs to be related to the delivery of pets, or is parcel distribution a different service

Line 28-29 – needs to be clear that the total number of access is 16 as you are counting probable as cases – I sound this section confusing - not sure why you are including the 7 non-cases who required medical services if not related to Q fever illness

Line 39 – needs to be clear the dust which was identified as same genotype a previous studies in that region was obtained from the pet holding site – it’s confusing the way it is written

Line 40-41 - the authors indicate that the source of the outbreak is transport of small ruminants which makes the title of the manuscript misleading unless the miniature goats are pets? – Or are they livestock/ruminants transported to farms?

Line 43 – miniature goats was the source of the outbreak yet the serology was negative – this is confusing and needs to be clarified, needs to link with the positive dust results

Line 44 – need more explanation or clarify what you mean by “Q fever was included among the occupational-associated health risks” – included in what?

Introduction

Line 51 – my understanding is that Q is not distributed worldwide – can this be checked please, a reference is also required

Line 53 - 55 – needs reference

Line 55 – what type of pets and wild species?

Line 57 – what are the animal reservoirs?

Line 64 – need to add “that” – “recent outbreaks that occurred”

Information about the incubation period is missing and should be added – this will help the reader to understand the epidemic curve and time of exposure ie. transportation of goats - to onset of illness

Material and Methods

Line 74 – is the courier company specific to pet transportation?

Line 84 – what type of urgent parcel delivery services – pets?

Line 89 – is delivery of pets to households, farms etc – need to be specific

Line 94 – what is the distance from local platform to Madrid and destination – is the route through country areas, or regions noted for livestock farming –? Possible exposure associated with windborne spread

Line 122-123 –what information was elicited from questionnaire – other risk factors or exposure information collected? How were participants recruited – eg. on-line survey, interview – this section needs more information – what was the time period

Line 124 – was the group of workers the 64 who completed the questionnaire? Please make this clearer

Line 139 – explain why the investigation was 3 weeks before – this is why you should specify the incubation period in the Introduction. What is meant by “suspect deliveries” When were the animals investigated – how long after, how were the located – had they already been delivered to their destination?

Line 144 -145 – here you should briefly how and where the environmental samples were taken

Line 155 – need to make it clear that the case definition included laboratory confirmed and probable cases, why did the authors include doubtful laboratory results?

Line 159 – here you mention personnel risk factors – but in the methods for data collection you don’t describe what information is collected - see my earlier comment on this

Results

Line 184-187 – I’m confused by the counts – in the Abstract 108 workers were exposed, yet in this section 180 workers were exposed. I’m also confused by 64 of the remaining 144 workers agreed to complete the questionnaire – where does 108 in the Abstract come from?

Line 87 – age group 6 to 20 is very large – is it possible to narrow the age group – is it more likely that cases were notified in children – this makes a difference when thinking about source of infection and risk factors

Line 88 – were the 42 cases reported during 2013 linked to an outbreak – or were they sporadic cases – outbreak cases will have a different relationship in that climate conditions may be different for those with a common exposure compared to sporadic cases with no identified source of infection

Line 90 – no mention of dust previously – see my earlier comments (line 68)

Line 94-95 – describe what is meant by “normal weather conditions”. Also state that you are referring to Table 2. Also explain wat is meant by “dust hovering”.

Line 96 – need explanation for dust hovering originated “inside” and “outside” the cities

Line 97 – state that you referring to Table 2. Also better to include the CI and not the p value

Line 234-235 – what I meant by “delivery considered of risk”

Line 262 – why is human vaccination not considered or discussed as a control measure

Discussion

Line 307 – was testing done on the flock of origin – how was trackback conducted and how many tested?

Parts of the Discussion was confusing to read and lacked clarity – eg. Line 302

I find it odd that there is mention of Q fever human vaccination as a control measure. In the last paragraph the authors discuss that a One Health approach is needed, yet there is no discussion of what One Health is and why it would be an effective approach.

Table 1

I found this table difficult to read and follow

Figure 1

Title missing for y-axis

6. PLOS authors have the option to publish the peer review history of their article (what does this mean?). If published, this will include your full peer review and any attached files.

Reviewer #1: No

Reviewer #2: No

Reviewer #3: No

---

## [Author Response · Author response to Decision Letter 0]

18 Oct 2019

PONE-D-19-18129

A Q fever outbreak associated to courier transport of pets

PLOS ONE

Editor Comments:

I invited and received three reviews for your manuscript. All reviews raised some substantial concerns about your manuscript as it currently stands. I read through their comments and found that they coincided on several points, and that their reviews were uniformly solid and detailed. I read the manuscript myself, and I must say that I coincide with the reviewers' points entirely. As such, I would recommend “major revision”. I would kindly ask you to go through all comments raised by each reviewer and address them properly before sending a revised version of this manuscript. Please check all PLOS ONE style requirements available via https://journals.plos.org/plosone/s/submission-guidelines before submitting the revised version.

AU: Thank you for giving us the opportunity to improve the manuscript. We have done the changes and modifications suggested by the reviewers, and we have answered the comments and questions point-by-point (see below).

Journal Requirements:

AU: We have followed the journal instructions and we think now the manuscript fits the style requirements of PLOS ONE, including file naming.

2. In your Methods, please state the volume of the blood samples collected for use in your study."

AU: In the revised manuscript we now state that tubes of 4mL were used for blood extraction (line 152).

In your revised cover letter, please address the following prompts

a)If there are ethical or legal restrictions on sharing a de-identified data set, please explain them in detail (e.g., data contain potentially identifying or sensitive patient information) and who has imposed them (e.g., an ethics committee). Please also provide contact information for a data access committee, ethics committee, or other institutional body to which data requests may be sent.

AU: The investigation of an outbreak is under the responsibility of the competent authority, which in this case is the Epidemiology Service of the Department of Public Health of the Basque Government. At the time of conducting the investigation of the outbreak described in this work, the written consent of the workers was requested for blood sampling and for conducting the surveys. Data derived from the questionnaires (including personal and clinical data) and the laboratory analyses results are available upon request to Eva Alonso from Public Health Department from the Basque Government (tepidebi-san@euskadi.eus).

Reviewers' comments:

Reviewer's Responses to Questions

Comments to the Author

1. Is the manuscript technically sound, and do the data support the conclusions?

Reviewer #1: Yes

Reviewer #2: Partly

Reviewer #3: No

2. Has the statistical analysis been performed appropriately and rigorously? 

Reviewer #1: I Don't Know

Reviewer #2: Yes

Reviewer #3: Yes

3. Have the authors made all data underlying the findings in their manuscript fully available?

Reviewer #1: Yes

Reviewer #2: Yes

Reviewer #3: Yes

4. Is the manuscript presented in an intelligible fashion and written in standard English?

Reviewer #1: No

Reviewer #2: Yes

Reviewer #3: No

5. Review Comments to the Author

Reviewer #1: Although I believe that the authors have investigated the outbreak with care, I do not believe that the article has any new or illuminating insights. It is not at all surprising that an outbreak of Q fever occurred in a business that transported goats and sheep. The association of Q fever with these ruminants was established decades ago, and the ability of Coxiella burnetii to survive in dust for long periods is also well-known. The clinical features of disease in the infected patients in this study are consistent with previous reports, with no novel elements noted.

AU: Yes, we understand perfectly the reviewer’s concern regarding novelty. However, the purpose of this article is to give an insight into the need to address the importance of testing the health status of pets before transport and highlight the important but generally neglected zoonotic risks associated to Coxiella when transporting pets by courier transport companies. In addition, this article highlights the difficulty of investigating Q fever outbreaks. By the time the first cases appear and suspicion of an outbreak is considered, in the absence of normalized action protocols, decisions ought to be taken on the spot based on the the oncoming results as they are being obtained. Regarding the clinical features associated to the affected patients, it is interesting to see how in northern Spain, Q fever develops with pneumonia rather than with liver disesase as is the case in other regions. The association of this case with the MST13 genotype is another important finding. This information is also of interest since there are not many studies in Spain that relate clinical forms with genotypes.

Reviewer #2: AUTHORS

Manuscript Number: PONE-D-19-18129

Manuscript Title: A Q fever outbreak associated to courier transport of pets

This is an outbreak report of Q fever in an express transport company linked to the transport of caprine, that had epidemic characteristics. It is an interesting and valid text that has an impact on the Public Health approach of this zoonosis. However I have major concerns that need to be addressed before acceptance.

1. Line 78: “possible cases of Q fever in two workers of the Araba branch of the same company who lived in Castilla-León.” What do you mean by Bizkaia and Araba branches? Are these cities? How far apart? How is this connected to Castilla-Leon? I believe this sentence would gain if rephrased for clarification.

AU: The reviewer is right, we talk about Bizkaia, Araba and Castilla-León, without explaining that they are provinces or regions in Spain. Now in the Case Presentation section of the revised manuscript there is a brief description of the location of the regions or provinces affected (lines 76-77).

2. Line 144: “masks). Later on, on July 22, a specialized” delete “on”

AU: ‘on’ has been deleted (line 124)

3. No environmental samples were taken from Araba and no link was found between these two branches that could justify a common cause (for example, same animal provider), other than occurring on the same dates. Additionally, molecular evidence exists only from Bizkaia and cannot confirm identic origins with Araba. Authors need to clarify why they considered this the same (multicentric?) outbreak.

AU: When the Epidemiology Service learned about the outbreak (August 3), a month had passed since the first cases with Q fever symptoms, and the company had already cleaned and disinfected the premises located both in Bizkaia and Araba (22 July). When environmental sampling was carried out on September 15, 2 months after the beginning of the outbreak, sampling efforts were focused on the pet holding site and surrounding areas of the platform of Bizkaia, because clinical and serological results indicated that the origin was there. It is true that Araba's facilities should have also been sampled, but since human cases in Araba platform ocurred soon after those in Bizkaia, everything pointed to Bizkaia as source of the outbreak. Therefore, and considering the period of time that had already lapsed since the onset of the outbreak, additional samplings were not considered, and the association with the Bizkaia branch was based on epidemiological data only. 

4. Authors suspect that two miniature goats travelling on June 28 from Bizkaia to Orense were the source of infection: reception at the Bizkaia platform – Araba platform (animals did not leave the van) – central headquarters at Madrid, where animals were transferred to another vehicle. How did these two animals shed C. burnettii in Araba (even not being handled), but in Madrid, where they were transferred and handled, no humans cases were reported? Although possible this is not likely, hence I would suggest to discuss these drawbacks.

AU: Yes, this is not easy to explain. One possibility, as stated in the discussion, might be that travel containers, transport vans or workers travelling from Bizkaia to Araba could have acted as vehicles for C. burnetii contamination of the Araba platform. In Madrid, since clinical symptoms compatible with Q fever were not reported by the workers, seroconversion was not assessed. In fact, no serological tests or epidemiological surveys were conducted in Madrid, and therefore infection status of the workers at Madrid was not investigated. 

Reviewer #3: “A Q fever outbreak associated to courier transport of pets” (PONE-D-19-18129) for PLOS ONE

Reviewer’s comments

This manuscript represents a detailed account of an outbreak investigation associated with transportation of pets via a courier transport service. While it is an important study, there are considerable shortcomings related to lack of detail in the methods concerning information collected in the questionnaire, participant recruitment, the counting of cases and non-cases is recorded differently in the abstract and in results, and there is lack of clarity in some sections.

AU: There was a mistake in the Abstract, and instead of 108 it should say 180 workers. This has now been changed in the revised manuscript (line 28). After gathering information from the company and all the organizations involved in the service, a total of 180 workers were identified as exposed. All of them were informed of the situation and the associated risks but for the reasons explained (residence in other communities, holiday periods, or lack of time, etc.) only 64 of them agreed to take part in the epidemiological study 

The questionnaire included questions related to patient, disease and occupational data, exposure risk outside the company, and personal risk associated data. The questionnaire used in the epidemiological investigation has been included in the manuscript as Supporting information (S1 Appendix) (line 136, line 449).

Title

The title is misleading as it indicates that the outbreak was associated with pets when in fact the source of infection was the carrying of goats – transportation of livestock/ruminants has a different risk to carrying domestic pets.

AU: The courier company has a Pet Transportation Service and the most important limit to provide transport service and therefore consider the animal a pet is the weight of the animal. They transport mainly dogs and cats, and eventually other pets, such as birds, rabbits, or miniature goats, like in this study. Although miniature goats are ruminants, they are most commonly kept as pets. As this Q fever outbreak took place in a transport company of these characteristics we thought it was interesting to reflect it in the title. Therfore, we would rather keep the title as it is.

Abstract

Line 26 – I’m confused as about the term parcel distribution – this needs to be related to the delivery of pets, or is parcel distribution a different service

AU: The term parcel distribution refers to the selection and distribution of the consignment according to destination. The same is done with animal cages, which have special storage places in the platforms. 

Line 28-29 – needs to be clear that the total number of access is 16 as you are counting probable as cases – I sound this section confusing - not sure why you are including the 7 non-cases who required medical services if not related to Q fever illness

AU: As soon as the company suspected of the possibility of infection among the workers, they were encouraged to visit their doctor or nearest hospital in case they had any symptom (fever, pneumonia, etc ...). Seven workers that participated in the questionnaire and had compatible symptoms (including one case of pneumonia) in the outbreak period did not meet the laboratory criteria of Q fever, and they were thus considered as Non-case. To avoid misunderstandings we have deleted this information from the Abstract and further explained it in the results section (lines 210-212) and Table 1.

Line 39 – needs to be clear the dust which was identified as same genotype a previous studies in that region was obtained from the pet holding site – it’s confusing the way it is written

AU: The sentence has been rephrased (line 41-42).

Line 40-41 - the authors indicate that the source of the outbreak is transport of small ruminants which makes the title of the manuscript misleading unless the miniature goats are pets? – Or are they livestock/ruminants transported to farms?

AU: As comented above, the courier company has a Pet Transportation Service with limit on the weight of the transported animals. They transport mainly dogs and cats, and eventually other pets, which include miniature goats. Miniature goats are frequently kept as pets. As the outbreak of Q fever took place in a courier company that transports miniature goats in the same manner that conventional pets are transported, we found it interesting to reflect it in the title.

Line 43 – miniature goats was the source of the outbreak yet the serology was negative – this is confusing and needs to be clarified, needs to link with the positive dust results

AU: As explained in the manuscript, the month before the onset of the Q fever outbreak there were 2 shipments of miniature goats, but only one could be serologically investigated. Being this one negative, it is likely that the outbreak originated in the other shipment of goats that took place in mid-June. Unfortunately, this is only a hypothesis that could not be confirmed. The implication of goats in other recent outbreaks of Q fever in the Basque Country, and the fact that the genotype found in this outbreak is commonly found in goats (SNP1-MST-13), led us to think that the cause of the outbreak was the transport of miniature goats, since due to weight restriction, the company does not transport conventional goats. 

In the revised manuscript, explicit reference to the miniature goats as the source of the outbreak has been deleted, and this is further explained in the body of the manuscript.

Line 44 – need more explanation or clarify what you mean by “Q fever was included among the occupational-associated health risks” – included in what?

AU: The company performs occupational health risk assessments as part of their Safety and Health Program. Before the outbreak, Q fever was not considered as an occupational health risk for the workers; after the outbreak, it has been included. This situation is further explained in the Discusion (lines 353-355, lines 369-370). 

Introduction

Line 51 – my understanding is that Q is not distributed worldwide – can this be checked please, a reference is also required

AU: Q fever has worldwide distribution and it has been found in all those countries where its presence has been investigated, with the exception of New Zealand. A reference has been added to support the sentence (line 53).

Line 53 - 55 – needs reference

AU: A reference has been added (line 58)

Line 55 – what type of pets and wild species?

AU: Two new references have been added. One to indicate that C. burnetii also infects pets, such as cats and dogs. Regarding wild animals, there are many species where C. burnetii DNA has been detected and we have selected one review which compiles many studies performed in wildlife.

Line 57 – what are the animal reservoirs?

AU: Animal reservoirs are those that maintain infection active and shed C. burnetii by different routes to the environment. We have added this explanation in the new version of the manuscript (lines 60-61).

Line 64 – need to add “that” – “recent outbreaks that occurred”

AU: Added (line 67)

Information about the incubation period is missing and should be added – this will help the reader to understand the epidemic curve and time of exposure ie. transportation of goats - to onset of illness

AU: The incubation period of Q fever is between 2 and 3 weeks. This information has been added in the text (lines 128-129)

Material and Methods

Line 74 – is the courier company specific to pet transportation?

Line 84 – what type of urgent parcel delivery services – pets?

AU: It is a courier company that in addition to urgent transport of parcels it is also authorized to transports animals, the only restriction being a maximum weight of 80 kg. Animals transported are mostly dogs and cats. When transporting animals, they arrive at destination within 24 hours after collection. 

Line 89 – is delivery of pets to households, farms etc – need to be specific

AU: It is not a conventional transport of farm animals; animals are individually transported between particulars. If the sender or recipient has a farm, this information is not registered by the courier company.

Line 94 – what is the distance from local platform to Madrid and destination – is the route through country areas, or regions noted for livestock farming –? Possible exposure associated with windborne spread

AU: The distance between the 2 platforms of Bizkaia and Araba is 60 km (indicated now in lines 83-84), and 361 km from the platform of Araba to Madrid (indicated now in line 245). The roads used are main highways (AP-1, A-1). The presence of farms along the route cannot be ruled out, and neither that the workers exposure had been associated with windborne spread of C. burnetii. However, the succession of clinical cases and the laboratory results, along with the epidemiological data derived from the questionnaire, suggest that exposure occurred at the platform of Bizkaia where most of the cases concentrated.

Line 122-123 –what information was elicited from questionnaire – other risk factors or exposure information collected? How were participants recruited – eg. on-line survey, interview – this section needs more information – what was the time period

AU: The translated questionnaire is now presented as supplementary data(S1 Appendix). Two on-site meetings were held, one at each platform, to inform the workers. During those meetings, blood samples were collected and workers answered the questionnaire (lines 132-137). Several workers did not attend these meetings but the Epidemiological Surveillance Unit phoned them and the workers completed the questionnaire by telephone and were asked to attend the health prevention service of the mutual insurance company to provide a blood sample for serological testing. 

Line 124 – was the group of workers the 64 who completed the questionnaire? Please make this clearer.

AU: Yes, 64 workers filled the questionnaire and also gave authorization for blood extraction.

Line 139 – explain why the investigation was 3 weeks before – this is why you should specify the incubation period in the Introduction. What is meant by “suspect deliveries” When were the animals investigated – how long after, how were the located – had they already been delivered to their destination?

AU: Now in the revised manuscript the incubation period of Q fever is mentioned (lines 128-129). Taking into account the incubation period of 2 to 3 weeks and the date of onset of symptoms of the first cases, the exposure period was established from June 1 until August 8. 

With the term ‘suspect deliveries’ we referred to the deliveries of ruminants that took place in this period and were therefore considered as possible source of the outbreak. There were two deliveries under suspicion but only one could be investigated at origin (August 23) and destiny (August 25), three weeks after the Epidemiological Surveillance Unit was informed of the Q fever cases. The term ‘suspect deliveries’ has been now deleted.

Line 144 -145 – here you should briefly how and where the environmental samples were taken

AU: Two aerosol samples were taken with an air sampler (Sartorius) inside the platform of Bizkaia. Dust samples were taken with cotton swabs in areas with accumulated dust in pet holding facilities and other places in the courier premises, giving a total of 19 surface dust samples analysed (lines 156-160). These samples were taken from the different surfaces indicated in Fig 2.

Line 155 – need to make it clear that the case definition included laboratory confirmed and probable cases, why did the authors include doubtful laboratory results?

AU: As indicated in the “Laboratory methods” section, “laboratory results were considered doubtful when only one blood sample was available and an indirect immunofluorescence antibody test (IFAT) titers for phase II IgG were >1/128 and for phase II IgM of >1/256.” In the context of an outbreak, the cases without symptoms but with a single positive serology are considered as probable cases so that they can be followed up to monitor a possible evolution towards chronic Q fever. 

Line 159 – here you mention personnel risk factors – but in the methods for data collection you don’t describe what information is collected - see my earlier comment on this

AU: The information included in the questionnaire has been added as Supporting information (S1 Appendix) (lines 448-449).

Results

Line 184-187 – I’m confused by the counts – in the Abstract 108 workers were exposed, yet in this section 180 workers were exposed. I’m also confused by 64 of the remaining 144 workers agreed to complete the questionnaire – where does 108 in the Abstract come from?

AU: As explained above there was a mistake in the abstract. Instead of 108, the total number of workers is 180. We have modified the text (line 28).

Line 87 – age group 6 to 20 is very large – is it possible to narrow the age group – is it more likely that cases were notified in children – this makes a difference when thinking about source of infection and risk factors

AU: We think that this and the next 5 comments are not related to this manuscript (No children were investigated in this study).

Line 88– were the 42 cases reported during 2013 linked to an outbreak – or were they sporadic cases – outbreak cases will have a different relationship in that climate conditions may be different for those with a common exposure compared to sporadic cases with no identified source of infection

AU: This comment is not related to this manuscript

Line 90 – no mention of dust previously – see my earlier comments (line 68)

AU: We cannot find in the manuscript any reference to the earlier comments of the reviewer 

Line 94-95 – describe what is meant by “normal weather conditions”. Also state that you are referring to Table 2. Also explain wat is meant by “dust hovering”.

AU: Table 2 does not exist. We do not mention in the manuscript the terms “normal weather conditions” or “dust hovering”.

Line 96 – need explanation for dust hovering originated “inside” and “outside” the cities

AU: This sentence does not appear in the text

Line 97 – state that you referring to Table 2. Also better to include the CI and not the p value

AU: Table 2 does not exist

Line 234-235 – what I meant by “delivery considered of risk”

AU: This has been rephrased (lines 241-242, 251) 

Line 262 – why is human vaccination not considered or discussed as a control measure

AU: Human vaccination has been used in Australia in groups of risk. This vaccine can induce local reactions, and patients should be evaluated with a cutaneous test for Q fever before vaccination to avoid severe side effects. At present, the use of this vaccine in humans has not been considered in Spain.

Discussion

Line 307 – was testing done on the flock of origin – how was trackback conducted and how many tested?

AU: The flock of origin had 5 miniature goats and 7 sheep. All animals were analysed by ELISA test giving negative results (lines 248-249).

Parts of the Discussion was confusing to read and lacked clarity – eg. Line 302

AU: This paragraph has been modified according the comments of the reviewer and we think now this part of the discussion is clear (lines 313-315).

I find it odd that there is mention of Q fever human vaccination as a control measure. In the last paragraph the authors discuss that a One Health approach is needed, yet there is no discussion of what One Health is and why it would be an effective approach.

AU: As stated above, the use of this vaccine in humans has not been considered in our country, which is considered a Q fever endemic area. The Australian vaccine, Q-VAX, is a whole-cell reactogenic vaccine that causes severe local and general reactions in persons who have been exposed to C. burnetii before. Negative serology, followed by negative skin testing is necessary before vaccination to avoid severe reactions. This vaccine is used for risk groups in Australia but it is not licensed or used in any other countries.The study of Q fever outbreaks is a clear example of teamwork in which public health, animal health and environmental health experts team together to establish a protocol of investigation, identify the source of infection and implement control measures to avoid risk for human population (lines 363-366).

Table 1

I found this table difficult to read and follow

AU: We have suppressed one column, in order to make the read of data easier. 

Figure 1

Title missing for y-axis

AU: Yes, it is true. When Power Point file was transformed in a ‘tif’ format, the title of y-axis was missed. Now, in the new versión the tittle appear as ‘no. cases’

---

## [Decision Letter · Decision Letter 1]

8 Nov 2019

A Q fever outbreak associated to courier transport of pets

PONE-D-19-18129R1

Dear Dr. García-Pérez,

We are pleased to inform you that your manuscript, "A Q fever outbreak associated to courier transport of pets" (PONE-D-19-18129R1), has been judged scientifically suitable for publication and will be formally accepted for publication once it complies with all outstanding technical requirements.

With kind regards,

Abdallah M. Samy, PhD

Academic Editor

PLOS ONE

---

## [Editor Report · Acceptance letter]

13 Nov 2019

PONE-D-19-18129R1 

A Q fever outbreak associated to courier transport of pets 

Dear Dr. García-Pérez:

I am pleased to inform you that your manuscript has been deemed suitable for publication in PLOS ONE. Congratulations! Your manuscript is now with our production department. 

With kind regards,

on behalf of

Dr. Abdallah M. Samy 

Academic Editor

PLOS ONE